# Elevated Lactate Dehydrogenase Has Prognostic Relevance in Treatment-Naïve Patients Affected by Chronic Lymphocytic Leukemia with Trisomy 12

**DOI:** 10.3390/cancers11070896

**Published:** 2019-06-26

**Authors:** Francesco Autore, Paolo Strati, Idanna Innocenti, Francesco Corrente, Livio Trentin, Agostino Cortelezzi, Carlo Visco, Marta Coscia, Antonio Cuneo, Alessandro Gozzetti, Francesca Romana Mauro, Anna Maria Frustaci, Massimo Gentile, Fortunato Morabito, Stefano Molica, Paolo Falcucci, Giovanni D’Arena, Roberta Murru, Donatella Vincelli, Dimitar G Efremov, Antonietta Ferretti, Gian Matteo Rigolin, Candida Vitale, Maria Chiara Tisi, Gianluigi Reda, Andrea Visentin, Simona Sica, Robin Foà, Alessandra Ferrajoli, Luca Laurenti

**Affiliations:** 1Institute of Hematology, Fondazione Policlinico Universitario A. Gemelli IRCCS, 00168 Roma, Italy; 2Departments of Leukemia, MD Anderson Cancer Centre, Houston, TX 77030, USA; 3Hematology and Clinical Immunology Unit, Department of Medicine, Università di Padova, 35122 Padova, Italy; 4Hematology Unit, IRCCS Ca’ Granda Policlinico-Università degli Studi, 55031 Milano, Italy; 5Division of Hematology, Ospedale San Bortolo di Vicenza, 36100 Vicenza, Italy; 6Department of Molecular Biotechnology and Health Sciences, University of Torino, 10126 Torino, Italy; 7Division of Hematology, A.O.U. Città della Salute e della Scienza di Torino, 10126 Torino, Italy; 8Hematology section, Department of Medical Sciences, Azienda Ospedaliero Universitaria Arcispedale S. Anna, 44124 Ferrara, Italy; 9Hematology Unit, Azienda Ospedaliera Universitaria Senese, 53100 Siena, Italy; 10Hematology division, Policlinico Umberto I, Università Sapienza, 00161 Roma, Italy; 11Department of Hematology, Ospedale Niguarda, 20162 Milano, Italy; 12Biothecnology Research Unit, Azienda Ospedaliera di Cosenza, 87100 Cosenza, Italy; 13Department of Hematology-Oncology, Ospedale Pugliese-Ciacco, 88100 Catanzaro, Italy; 14Division of Hematology, Ospedale Belcolle, 01100 Viterbo, Italy; 15Hematology and Stem cell Transplantation Unit, IRCCS Centro di Riferimento Oncologico della Basilicata, 85028 Rionero in Vulture, Italy; 16Hematology and Stem Cell Transplantation Unit, Ospedale A. Businco, 09121 Cagliari, Italy; 17Department of Hematology, Azienda Ospedaliera Bianchi-Melacrino-Morelli, 89124 Reggio Calabria, Italy; 18Molecular Hematology, International Centre for Genetic Engineering and Biotechnology, 34149 Trieste, Italy; 19Institute of Hematology, Università Cattolica del Sacro Cuore, 00168 Roma, Italy

**Keywords:** CLL, trisomy 12, LDH, prognosis

## Abstract

Chronic Lymphocytic Leukemia (CLL) patients with +12 have been reported to have specific clinical and biologic features. We performed an analysis of the association between demographic; clinical; laboratory; biologic features and outcome in CLL patients with +12 to identify parameters predictive of disease progression; time to treatment; and survival. The study included 487 treatment-naive CLL patients with +12 from 15 academic centers; diagnosed between January 2000 and July 2016; and 816 treatment-naïve patients with absence of Fluorescence In Situ Hybridization (FISH) abnormalities. A cohort of 250 patients with +12 CLL followed at a single US institution was used for external validation. In patients with +12; parameters associated with worse prognosis in the multivariate model were high Lactate DeHydrogenase (LDH) and β-2-microglobulin and unmutated immunoglobulin heavy-chain variable region gene (*IGHV*). CLL patients with +12 and high LDH levels showed a shorter Progression-Free-Survival (PFS) (30 months vs. 65 months; *p* < 0.001), Treatment-Free-Survival (TFS) (33 months vs. 69 months; *p* < 0.001), Overall Survival (OS) (131 months vs. 181 months; *p* < 0.001) and greater CLL-related mortality (29% vs. 11% at 10 years; *p* < 0.001) when compared with +12 CLL patients with normal LDH levels. The same differences were observed in the validation cohort. These data suggest that serum LDH levels can predict PFS; TFS; OS and CLL-specific survival in CLL patients with +12.

## 1. Introduction

The clinical course of chronic lymphocytic leukemia (CLL) is very heterogeneous and the identification of prognostic and predictive factors for CLL is of great relevance and a field of active investigation [1,2]. During the last decade several laboratory biomarkers have been identified as being correlated with outcomes for risk assessment [3,4,5,6]. The analysis of aberrant chromosomal regions with specific DNA probes by fluorescence in situ hybridization (FISH) resulted in the detection of clonal aberrations and the main recurrent chromosomal abnormalities (del13q, +12, del11q and del17p) define different personal genetic profiling subgroups in CLL [7,8]. 

Trisomy 12 is the second most frequent cytogenetic abnormality identified by FISH in patients with CLL [7]. It presents as an isolated aberration in about 70% of cases and when it is associated with additional chromosomal abnormalities portends a poor prognosis [7,8,9,10,11].

CLL patients with +12 have unique morphologic and immunophenotypic characteristics [12].

CLL cells with +12 commonly have an atypical morphology, defined as the presence of cleaved nuclei and/or lymphoplasmacytoid features in more than 15% of cells [13,14,15,16]. When analyzing their immunophenotype, cases with +12, in comparison with cases with a normal FISH, show a significantly higher expression of CD19, CD22, CD20, CD79b, CD24, CD27, CD38, CD49d, sIgM, sIgk and sIgλ and a lower expression of CD43 [9]. 

CLL patients with +12 show a greater percentage of unmutated immunoglobulin heavy-chain variable region gene (*IGHV*) cases (54%) versus those with del13q and normal karyotype (37% and 31%, respectively) [14,15,16,17]. *IGHV* studies reported a significantly more frequent expression of stereotyped B-cell receptors compared to patients with CLL and no +12 (44% vs 27% respectively), with a higher prevalence of the *IGHV* 4–39 gene, particularly in cases that later developed Richter’s syndrome (RS) [17,18,19]. The role of the *IGHV* mutational status in predicting the clinical course of CLL patients with +12 has been well investigated by Bulian et al. [20]: it proved to be the sole prognostic factor able to stratify overall survival (OS) and time-to-first treatment (TTFT) in +12 CLL.

CLL patients with +12 rarely have TP53 mutations or acquire them over time [6,21]. On the contrary, *NOTCH1* mutations are very frequent in +12 CLL patients and are detected in 30–40% of the cases [22,23,24].

In the literature there are few reports that describe in detail the clinical features of CLL cases with +12 [17,25,26,27]: the two largest series were reported by Marin et al., (289 patients) [26] and Strati et al., (250 patients) [27]. We now present a large series of treatment-naive +12 CLL patients and correlate the association between demographic, clinical, laboratory, and biologic features and clinical outcomes. 

## 2. Results

### 2.1. Patients’ Characteristics

Four-hundred and eighty-seven patients with CLL and +12 and 816 patients with negative FISH were included in the study. +12 patients had a median age at diagnosis of 65.5 years (range 33–90) with a male/female (M:F) ratio of 1.68 (305 males, 63%; and 182 females, 37%). Negative FISH patients had a median age at diagnosis of 63.0 years (range 27–93) and a male/female M:F ratio of 1.58 (500 males, 61%; and 316 females, 39%). All clinical, laboratory, and biologic characteristics at diagnosis are presented in Table 1. 

Patients with +12 had significantly higher levels of lactate dehydrogenase (LDH) and β-2-microglobulin; more frequently expressed ZAP70, CD38, and CD49d (*p* < 0.001); and more frequently had an unmutated *IGHV* status compared to the control group (*p* < 0.001). 

### 2.2. Outcomes

#### 2.2.1. Patients +12 CLL vs. Negative FISH CLL

Among the 487 CLL patients with +12, 311 (64%) progressed, 298 received treatment (61%), and 125 (26%) died, resulting in a median progression-free-survival (PFS) of 51 months (confidence interval (CI) 95%: 44 to 64), a median treatment-free-survival (TFS) of 59 months (CI 95%: 47 to 65) and a median OS of 170 months (CI 95%: 147 to 182). We then analysed data regarding the 816 CLL patients with negative FISH: 374 progressed (46%), 328 received treatment (40%), and 145 died (18%), resulting in a median PFS of 100 months (CI 95%: 81 to 118), a median TFS of 141 months (CI 95%: 112 to 160), and a median OS of 199 months (CI 95%: 196 to 210). When comparing the curves, we noted significantly shorter survival of +12 patients than FISH negative in terms of PFS (*p* < 0.001; Figure 1A), TFS (*p* < 0.001; Figure 1B) and OS (*p* < 0.001; Figure 1C). No differences were noted in the occurrence of Richter transformation and second cancers between the two groups.

We next performed analysis of all the investigated categorical and continuous variables to identify parameters associated with shorter survival in +12 CLL patients. In univariate analysis, Binet and Rai advanced stage, elevated LDH, and β-2-microglobulin levels and unmutated *IGHV* were associated with shorter PFS, TFS, and OS; ZAP70 positivity was associated with shorter TFS and OS; whereas older age and positivity for CD38 were associated only with shorter OS. In multivariate analysis, elevated LDH and β-2-microglobulin levels, unmutated *IGHV*, and Rai advanced stage maintained their association with PFS and TFS, whereas elevated LDH and β-2-microglobulin levels, unmutated *IGHV*, and older age showed significance for OS (Appendix A). 

We noticed that elevated LDH levels at diagnosis were more common in CLL patients with +12 compared to negative FISH patients (31% versus 14%, *p* < 0.001). These levels, as a categorical variable in which each centre declared their ranges and upper limits, were considered elevated if stable over the time. All CLL patients included in the analysis did not show lymphadenomegaly or symptoms suggestive of RS or haemolytic anaemia at the onset of the disease that could cause high LDH levels. 

#### 2.2.2. Characteristics of +12 and FISH-Negative CLL Patients Stratified according to LDH Levels

Since LDH is an inexpensive and routinely performed laboratory parameter, we evaluated the frequency of elevated levels of LDH at diagnosis in +12 CLL patients. Among patients with +12, 142 (31.4%) patients showed elevated LDH levels and 310 (68.6%) patients showed levels within normal range. The characteristics of the two subgroups are summarized in Table 2. 

The patients with +12 and high LDH levels progressed in 72% of the cases with a median PFS of 30 months (CI 95% 20–42) and needed treatment in 70% of the cases with a median TFS of 33 months (CI 95% 24–45). In comparison, the patients with +12 and normal LDH levels progressed in 59% of the cases with a median PFS of 65 months (CI 95% 58–73) and needed treatment in 55% of cases with a median TFS of 69 months (CI 95% 62–89). Regarding OS, 113 patients died among the +12 CLL patients, 47 (33%) of those with high LDH and 66 (21%) of those with normal LDH. The median OS was 131 months (CI 95% 97–167) in the patients with high LDH and 181 months (CI 95% 166–199) in the patients with normal LDH. Comparing the outcomes between the two subgroups, statistically significant shorter PFS (*p* < 0.001; Figure 2A), TFS (*p* < 0.001; Figure 2B), and OS (*p* < 0.001; Figure 2C) were observed in the patients with high LDH levels.

We then compared the causes of death in these patients. Thirty (64% of all the deaths) in the high LDH group and 23 (35%) in the normal LDH group were due to CLL. The median CLL-specific survival was 147 months in the high LDH group and 190 months in the normal LDH group with a different rate of CLL-related mortality at 10 years: 29% vs. 11% (Figure 3).

When performing multivariate analysis in CLL patients with +12 according to CLL-specific survival, the role of LDH was confirmed (*p* < 0.001, hazard ratio (HR) 3.78, CI 95% 1.73–8.26; Appendix A).

We then investigated whether the negative prognostic role of LDH observed in +12 CLL could be extended to CLL patients with negative FISH. Analyzing the same variables and using the same univariate model followed by a multivariate model for the significant variables, LDH had a role for PFS and TFS in univariate analysis but it was not significant in multivariate analysis in which Binet stage, CD38, β-2-microglobulin, and the *IGHV* status showed a significant role. With respect to OS and CLL-specific survival, LDH was not significant even in univariate analysis (Appendix A).

Patients with +12 and normal LDH levels showed no significant differences in survival compared to patients with negative FISH (*p* = 0.22 for OS and *p* = 0.61 for CLL-specific survival; Figure 4A,B) indicating that the difference in outcomes is dependent on the patients with +12 and elevated LDH.

#### 2.2.3. Validation Cohort

To validate the worse prognosis of +12 CLL patients with high LDH levels, the same analysis was performed on a population of 250 patients with +12 from a single US institution. Baseline patients’ characteristics at presentation are shown in Appendix A. This population was divided according to LDH levels available at diagnosis: the two subgroups of 104 and 145 patients with high LDH levels and normal LDH levels, respectively, are presented in Appendix A. Differences in the outcomes were found also in these subgroups: patients with high LDH levels showed a shorter median PFS (24 months vs. 55 months in patients with normal LDH levels, *p* < 0.001; Figure 5A), shorter median TFS (25 months vs. 58 months, *p* < 0.001; Figure 5B), and higher death rate (22% vs. 11% from all causes and 65% vs. 12% for CLL-related mortality). Also OS (92 months vs. 103 months; *p* = 0.012; Figure 5C) and CLL-specific survival (99 months vs. 128 months; *p* < 0.001; Figure 5D) were significantly shorter in the high LDH subgroup.

A univariate analysis was performed in this validation cohort, as in the multicenter cohort: Rai stage, LDH, ZAP70 and β-2-microglobulin resulted significant for PFS and TFS; age, LDH and β-2-microglobulin for OS; LDH and β-2-microglobulin for CLL-specific survival. When these variables were analysed in a multivariate model, LDH was the sole negative independent parameter for PFS, TFS, and CLL-specific survival; LDH and age were significant for OS (Appendix A).

## 3. Discussion

Our retrospective study confirmed that CLL patients showing +12 on FISH analysis have unique clinical and biologic features as reported in the literature [12,25]. The higher expression of CD38 observed in +12 by Athanasiadou et al. [14] was confirmed by our data with a rate of 51%; also the rates of CD49d and ZAP70 were higher in +12 CLL vs. negative FISH CLL (79% and 54% vs. 27% and 36%, respectively). Finally, the higher prevalence of unmutated *IGHV* in our series of +12 CLL patients (57%) confirmed previously published data [17].

In our series, CLL patients with +12 had a worse prognosis compared to CLL patients with negative FISH disease. The biomarkers associated with shorter PFS, TFS, OS in multivariate analysis were high LDH, unmutated *IGHV*, and elevated β-2-microglobulin. It resulted that Rai stage was significant only for shorter PFS and shorter TFS, and age only for shorter OS. 

Data about the correlation between the *IGHV* status and outcome in CLL are well known from the literature, including +12 patients [20]. Also in our cohort, high levels of β-2-microglobulin predicted a worse prognosis in terms of PFS, TFS, and OS, as reported in the literature. Advanced Rai stage had an impact on PFS and TFS and older age on OS, as expected.

Analyses about clinicopathologic risk categorization of untreated CLL patients aimed to find predictive outcomes variables: a parameter such as LDH was identified in a study as an independent predictor of TFS independently from FISH categories and the impact of LDH was greater for patients with unmutated *IGHV* [28], in another case LDH was not included among the variables for a comprehensive prognostic index [29], but our report is the first one about biomarker approaches on the relationship between +12, high LDH levels, and outcome using both a discovery and a validation series of a great size.

When we attempted to analyse the role of LDH by stratifying the +12 CLL cohort according to the LDH levels at diagnosis, patients with high LDH levels showed a significantly worse outcome in terms of PFS, TFS, OS, and CLL-specific survival both in univariate and multivariate analyses. None of the +12 CLL patients, including those with high LDH levels, showed signs or symptoms suggestive of RS at the onset of the disease.

Overall, our data are consistent with previous reports that +12 CLL patients have an intermediate risk of progression compared to the other FISH-defined prognostic subsets [7]. However, stratifying by LDH levels, it appeared clear that +12 CLL patients with high LDH levels showed a worse prognosis compared to +12 CLL patients with normal LDH levels, for whom prognosis was similar to that of patients with negative FISH.

In a wide validation cohort of 250 patients, the parameters which resulted significant were comparable to the ones of the multicenter population, confirming the predictive role of LDH: it resulted that it was the sole negative independent biomarker for PFS, TFS, and CLL-specific survival and it was significant for OS together with age. These data reinforced the impact of LDH in the +12 CLL population, which is marked by unique clinical and biological features that could explain a high rate of LDH levels above the limit, possibly linked also to atypical morphological characteristics of their cells. So it is suggested that LDH, an easily available individualized tool capable of predicting a worse PFS, TFS, and OS, should be taken into account in daily clinical practice.

It would be useful in future studies to investigate outcomes in the LDH-high and LDH-normal +12 CLL patients according to the treatment received, in particular after therapy with the novel targeted agents. Unfortunately, this was not possible in the current study because of the relatively small number of patients treated with targeted agents and short follow-up, precluding a meaningful statistical comparison. In addition, future studies should address the association between high LDH levels and NOTCH1 mutation, which has been detected in 30–40% of +12 cases. Such data were not available for our current study and may provide an explanation for the correlation between high LDH levels and poor prognosis in CLL patients with the +12 abnormality.

## 4. Materials and Methods

This is a retrospective observational study including treatment-naive CLL patients with +12 as an isolated aberration from 15 academic Italian centers, diagnosed between January 2000 and July 2016. A second cohort of patients who resulted negative for a FISH panel comprising the four common abnormalities, i.e., del13q, +12, del11q, and del17p, was collected from the same centers during the same study period. A previously published overlapping cohort of patients with +12 CLL followed at a single US institution was used for external validation but new and updated data were included [27]. 

CLL was diagnosed in all the patients according to the 2008 International Workshop on CLL (iwCLL) guidelines [30]. Patients were also screened at baseline with a direct antiglobulin test and radiological examinations and if other possible causes of LDH elevation such as signs or symptoms suggestive of RS or haemolytic anaemia were noted, these patients were excluded from the analysis. 

Demographic, clinical, and laboratory data were collected for each patient. Older age was defined as ≥65 years or older, advanced stage as Rai stage III–IV and Binet stage C. 

*IGHV* somatic mutation status and expression of CD38, ZAP70, CD49d, were performed in a standardized fashion by all involved centers, as previously described [31,32]. FISH analysis was performed on interphase nuclei of CLL cells of the peripheral blood and the panel included probes specific to TP53 (17p13.1), ATM (11q22.3), D13S319 (13q14.3), LAMP1 (13q34), and the centromeric region of chromosome 12 (12p11.1–q11) [33]. Disease progression, treatment initiation, and death were recorded to calculate PFS, TFS, and OS. 

The study was approved by the Ethics Committee of the ’Fondazione Policlinico Gemelli’ (Protocol No. 0028829/16; Date: 13th July 2016) and was conducted in accordance with the principles of the Declaration of Helsinki. The clinical and laboratory features were obtained by review of the medical records and all the data were centrally collected and analysed. 

PFS, TFS, and OS were calculated from the date of diagnosis to the date of progression, treatment, and death, respectively, or the date of last follow-up. We defined CLL-specific death as death secondary to progressive disease, RS, infections, and complications during the treatment or in patients with an active disease.

Normality distributions of all variables were tested by the Shapiro-Wilk and Shapiro-Francia tests. Chi-square test or Fisher’s exact test were used to compare categorical variables (age, gender, Binet and Rai stage, LDH and β-2-microglobulin levels, positivity for ZAP70, CD38 and CD49d, IGHV mutational status), while the Wilcoxon-Mann-Whitney test was applied for continuous variables (white blood and lymphocytes counts, haemoglobin and platelets levels, bone marrow infiltration). Data were summarized as medians, 25th and 75th percentiles. The Kaplan–Meier method was used for survival analyses, and the log-rank test was performed to compare patient subgroups. Univariate and multivariate Cox proportional hazards regression models were fit to assess associations between patients’ characteristics and survival times. The proportional hazards assumption was assessed using the method of Grambsch et al. [34]. Only variables in univariate Cox analysis with a *p* < 0.01 (or *p* < 0.05 for validation cohort) were added to the multivariate Cox regression model. *p* values lower than 0.05 were considered statistically significant and reported as two-sided. All statistics were carried out with the use of STATA/SE 12.0 for Windows.

## 5. Conclusions

From our multicenter study on 487 patients with +12 and 816 patients with negative FISH and from the analysis of our validation cohort on 250 patients with +12, it emerged that (1) CLL patients with +12 have a significantly higher prevalence of elevated LDH compared to patients without FISH abnormalities, (2) +12 CLL patients have a worse prognosis than negative FISH CLL patients, (3) +12 CLL patients with normal LDH levels have the same prognosis as that of negative FISH patients, and (4) +12 CLL patients with high LDH levels show a worse PFS, TFS, and OS than +12 CLL patients with normal LDH levels or negative FISH patients independently of LDH levels.

## Figures and Tables

**Figure 1 cancers-11-00896-f001:**
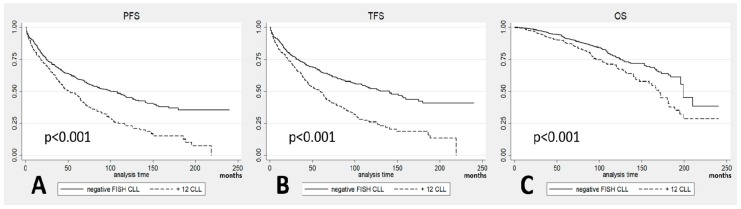
(**A**) Progression-free-survival (PFS) in patients with +12 chronic lymphocytic leukemia (CLL) vs. fluorescence in situ hybridization (FISH) negative CLL; (**B**) treatment-free-survival (TFS) in patients with +12 CLL vs. FISH negative CLL; (**C**) overall survival (OS) in patients with +12 CLL vs. FISH negative CLL.

**Figure 2 cancers-11-00896-f002:**
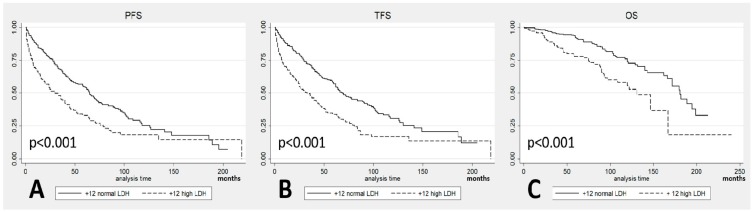
(**A**) Progression-free-survival (PFS) in patients with +12 stratified according to Lactate DeHydrogenase (LDH) levels; (**B**) Treatment-free-survival (TFS) in patients with +12 stratified according to LDH levels; (**C**) Overall survival (OS) in patients with +12 stratified according to LDH levels.

**Figure 3 cancers-11-00896-f003:**
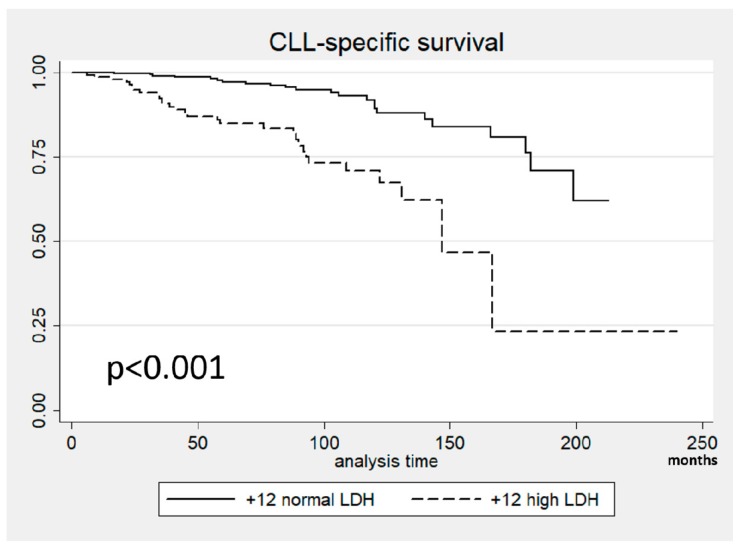
Chronic Lymphocytic Leukemia (CLL)-specific survival in patients with +12 stratified according to Lactate DeHydrogenase (LDH) levels.

**Figure 4 cancers-11-00896-f004:**
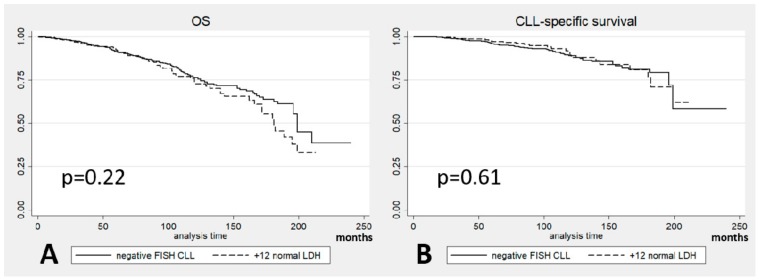
(**A**) Overall survival (OS) and (**B**) Chronic Lymphocytic Leukemia (CLL)-specific survival in +12 CLL patients with normal Lactate DeHydrogenase (LDH) levels vs. Fluorescence In Situ Hybridization (FISH) negative CLL patients.

**Figure 5 cancers-11-00896-f005:**
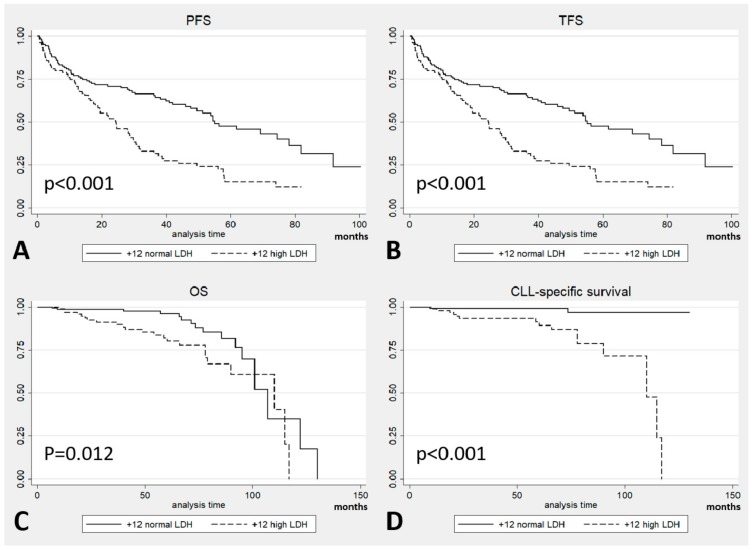
Outcomes in the validation cohort by Lactate DeHydrogenase (LDH) levels: (**A**) Progression-free-survival (PFS); (**B**) Treatment-free-survival (TFS); (**C**) Overall survival (OS); and (**D**) Chronic Lymphocytic Leukemia (CLL)-specific survival.

**Table 1 cancers-11-00896-t001:** Patients’ baseline characteristics at diagnosis.

		Patients with FISH +12 (487 Patients)	Patients with FISH Negative (816 Patients)	*p*
**Median age (range)**		65.5 (33–90)	63.0 (27–93)	0.002
**Gender M/F (ratio)**		305/182 (1.68)	500/316 (1.58)	ns
**Binet stage**	**A** **B** **C**	359 (73.7%) 104 (21.4%) 24 (4.9%)	663 (81.3%) 103 (12.6%) 50 (6.1%)	<0.001
**Rai stage**	**0** **I-II** **III-IV**	237 (48.7%) 225 (46.2%) 25 (5.1%)	500 (61.3%) 265 (32.4%) 51 (6.3%)	<0.001
**Palpable splenomegaly**		112 (23.0%)	135 (16.5%)	0.004
**Palpable hepatomegaly**		64 (13.1%)	66 (8.1%)	0.003
**Lymphadenopathies >5 cm**		243 (49.9%)	298 (36.5%)	<0.001
**White blood cells (mmc) (range)**		15,675 (11,675–24,600)	15,610 (11,900–22,610)	ns
**Lymphocytes peripheral blood (mmc) (range)**		10,190 (6625–18,965)	10,800 (7090–16,800)	ns
**Hemoglobin (g/dL) (range)**		13.7 (12.8–14.8)	13.9 (12.7–14.9)	ns
**Platelets (mmc) (range)**		193,000 (156,000–239,000)	200,000 (160,000–244,000)	ns
**LDH**	**Normal levels** **Above the limit**	310/452 (68.6%) 142/452 (31.4%)	662/767 (86.3%) 105/767 (13.7%)	<0.001
**Lymphocytes bone marrow (%)**		68 (50–80) in 152/487 pts	60 (40–79) in 197/816 pts	ns
**ZAP70 positive (≥20%)**		197/363 (54.3%)	241/676 (35.6%)	<0.001
**CD38 positive (≥30%)**		222/433 (51.3%)	213/768 (27.7%)	<0.001
**CD49d positive (≥30%)**		89/113 (78.8%)	54/201 (26.9%)	<0.001
**β-2-microglobulin**	**Normal levels** **Above the limit**	195/396 (49.2%) 201/396 (51.8%)	436/654 (66.7%) 218/654 (33.3%)	<0.001
***IGHV* mutational status**	**Mutated** **Unmutated**	164/384 (42.7%) 220/384 (57.3%)	430/679 (63.3%) 249/679 (36.7%)	0.001

FISH: Fluorescence In Situ Hybridization; M/F: male/female; LDH: Lactate DeHydrogenase; IGHV: immunoglobulin heavy-chain variable region gene.

**Table 2 cancers-11-00896-t002:** Baseline characteristics at diagnosis of CLL patients with +12 divided in two subgroups according to lactate dehydrogenase (LDH) levels.

		Patients with High LDH Levels (142 Patients)	Patients with Normal LDH Levels (310 Patients)	*p*
**Median age (range)**		65.5 (33–89)	65.5 (33–90)	ns
**Gender M/F (ratio)**		82/60 (1.37)	201/109 (1.84)	ns
**Binet stage**	**Stage A** **Stage B** **Stage C**	84 (59.1%) 45 (31.7%) 13 (9.2%)	249 (80.3%) 51 (16.5%) 10 (3.2%)	<0.001
**Rai stage**	**Stage 0** **Stage I–II** **Stage III–IV**	47 (33.1%) 81 (57.0%) 14 (9.9%)	168 (54.2%) 132 (42.6%) 10 (3.2%)	<0.001
**Palpable splenomegaly**		47 (33.1%)	58 (18.7%)	0.001
**Palpable hepatomegaly**		23 (16.2%)	34 (11.0%)	ns
**Lymphadenopathies >5 cm**		89 (62.7%)	141 (45.5%)	0.001
**White blood cells (mmc) (range)**		18,500 (12,800–32,370)	14,680 (11,100–21,100)	<0.001
**Lymphocytes peripheral blood (mmc) (range)**		12,344 (7811–24,175)	9245 (5990–15,000)	<0.001
**Hemoglobin (g/dL) (range)**		13.4 (12.2–14.6)	13.9 (13.0–15.0)	<0.001
**Platelets (mmc) (range)**		185,000 (153,000–235,000)	194,000 (159,000–238,000)	ns
**Lymphocytes bone marrow (%)**		70 (50–85) in 63/142 pts	60 (45–77) in 86/310 pts	0.03
**ZAP70 positive (≥20%)**		64/107 (59.8%)	117/231 (50.6%)	ns
**CD38 positive (≥30%)**		71/133 (53.4%)	132/267 (49.4%)	ns
**CD49d positive (≥30%)**		31/38 (81.6%)	58/75 (77.3%)	ns
**β-2-microglobulin**	**Normal levels** **Above the limit**	47/134 (35.1%) 87/134 (64.9%)	146/258 (56.6%) 112/258 (43.4%)	<0.001
***IGHV* mutational status**	**Mutated** **Unmutated**	40/117 (34.2%) 77/117 (65.8%)	113/238 (47.5%) 125/238 (52.5%)	0.017

M/F: male/female; LDH: Lactate DeHydrogenase; IGHV: immunoglobulin heavy-chain variable region gene.

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
