# Peer review of "Elevated Lactate Dehydrogenase Has Prognostic Relevance in Treatment-Naïve Patients Affected by Chronic Lymphocytic Leukemia with Trisomy 12"

_cancers, 2019, doi:10.3390/cancers11070896_

Round 1

Reviewer 1 Report

The authors analyse a cohort of patients with CLL and +12 to specifically identify prognostic variables applicable for this genetic subset.

Major issues:

Does this cohort overlap[ with the previously reported cohort from Starti (Ref 27)? If so, please explicitly state this and describe overlap.

The timepoint in the disease course (e.g. at diagnosis) when testing and biochemical variables were collected needs to be clearly stated.

What was the actual rate of RSA in the +12 cohort? It is somewhat unexpected that this was not greater than the “normal FISH” comparators.

The section of comparisons between +12 with elevated LDH and “normal FISH” cohort add little to the paper and can be omitted.

What treatment did these patients receive? Presumably most patients were NOT treated with targeted agents (Ibrutinib, venetoclax etc…). Given the increasing earlier utilisation of these agents, will the prognostic significance of LDH be retained with such novel treatments? Emerging data suggests this may be the case at least in the relapsed setting (Soumerai Lancet Haematol 2019) but this issue needs to be considered and discussed.

A major deficiency here is the absence of mutational data for NOTCH1 – can this gap be addressed?

Minor comments:

30 authors is a very large number for such a manuscript; please confirm all authors fulfil Vancouver criteria for inclusion?

The text is very “staccato” comprising multiple very short paragraphs, some just a single sentence,  with poor flow. Please reformat and aggregate text into more coherent paragraphs.

It is unexpected that all patients had a bone marrow biopsy performed at the time of diagnosis – please clarify whether this was the case, or whether bone marrow data is available only on a smaller subset.

Author Response

June 16, 2019

To: Danica Jevtic

Associate Editor,

Cancers

Dear Dr. Danica Jevtic,

We submit to your attention the revised version of our manuscript entitled “Elevated Lactate Dehydrogenase Has Prognostic Relevance in Treatment-Naïve Patients Affected by Chronic Lymphocytic Leukemia with Trisomy 12”.

We appreciate the time, dedication and constructive criticism each reviewer provided.

A point-by-point response to reviewers’ comments is outlined below.

Reviewer: 1

Comments and Suggestions for Authors

The authors analyse a cohort of patients with CLL and +12 to specifically identify prognostic variables applicable for this genetic subset.

Major issues:

Does this cohort overlap[ with the previously reported cohort from Starti (Ref 27)? If so, please explicitly state this and describe overlap.

ANSWER. The validation cohort presented in the manuscript is the same previously reported by Strati et al. This has been clarified in the Materials and Methods section. However, as compared to the previously published series, we would also like to note that the data have been updated and new variables, not included in the previous study, were investigated and added. 

The timepoint in the disease course (e.g. at diagnosis) when testing and biochemical variables were collected needs to be clearly stated.                                                                                                                  

ANSWER. Variables were collected at diagnosis. This has been clarified in the revised manuscript, adding it in the text and in the tables.

What was the actual rate of RSA in the +12 cohort? It is somewhat unexpected that this was not greater than the “normal FISH” comparators.

ANSWER. The rate of RS in the +12 cohort was 2.6%; it was greater but not statistically significant than the one in the ‘normal FISH’ cohort (1.9%).

The section of comparisons between +12 with elevated LDH and “normal FISH” cohort add little to the paper and can be omitted.

ANSWER. We agree with the reviewer and removed this session.

What treatment did these patients receive? Presumably most patients were NOT treated with targeted agents (Ibrutinib, venetoclax etc…). Given the increasing earlier utilisation of these agents, will the prognostic significance of LDH be retained with such novel treatments? Emerging data suggests this may be the case at least in the relapsed setting (Soumerai Lancet Haematol 2019) but this issue needs to be considered and discussed.

ANSWER. As data cut-off dates back to 2016, only few patients were treated with targeted agents. As such, unfortunately, it is not possible to run the requested analysis, which we agree would be highly relevant, and that we hope to prospectively perform in the future.

A major deficiency here is the absence of mutational data for NOTCH1 – can this gap be addressed?

ANSWER. We agree with the reviewer that this would be of great interest to the reader. Unfortunately, as explained in the text, we had mutational data for NOTCH1 for only a few patients, so this gap currently can not be addressed. However, we definitely aim to collect this prospectively for future analyses.

Minor comments:

30 authors is a very large number for such a manuscript; please confirm all authors fulfil Vancouver criteria for inclusion?

ANSWER. A high number of coauthors was needed to collect such a large cohort of patients from different centers. All authors fulfil Vancouver criteria.

The text is very “staccato” comprising multiple very short paragraphs, some just a single sentence, with poor flow. Please reformat and aggregate text into more coherent paragraphs.

ANSWER. We agree with the reviewer and reformatted the manuscript accordingly.

It is unexpected that all patients had a bone marrow biopsy performed at the time of diagnosis – please clarify whether this was the case, or whether bone marrow data is available only on a smaller subset.

ANSWER. A bone marrow biopsy was performed at baseline in 349/1303 (27%) patients. The rates of each subgroup have been added to the manuscript. 

We also performed general English editing and provided the correct email for one of the co-authors.

We hope that the revised version will be considered acceptable for publication in Cancers.

Please, do not hesitate to contact us should further modifications be required.

Sincerely,

Luca Laurenti

Reviewer 2 Report

In this manuscript Autore et al. performed a retrospective analysis of a large cohort of treatment-naïve +12 CLL patient data obtained from multiple Italian centers. They aimed to determine whether specific clinical features of +12 CLL patients were associated with a poor prognosis according to progression-free survival, treatment-free survival, overall survival and CLL-specific survival. Using Cox regression, multivariate analysis revealed that elevated beta-2-microglubulin and LDH were significant risk factors for poor prognosis in all four categories. Beta-2-microgobulin levels are known to be associated with CLL disease burden, and the authors focused on the novelty of identifying LDH as a prognostic indicator for +12 CLL patients. The authors analyzed clinical data from a separate American cohort, and similarly found that elevated LDH was associated with poor prognosis for +12 CLL patients. Surprisingly, they did not identify elevated LDH as a prognostic indicator for negative FISH CLL patients. Overall this manuscript is well written and would be of interest to the CLL field. The following comments might help improve its quality.

The axes of the Kaplan-Meier plot should have a more descriptive label (time is months).

In the Discussion, the authors to should speculate on the mechanism and relevance to the disease of high LDH: Without normal underlying signatures, why is LDH elevated? If elevated LDH is a poor prognosis indicator for +12 CLL, why is it not so for negative FISH CLL?

Is the Forest plot in Figure 4 necessary/appropriate. These plots are normally used to compare data from similar studies (studies examining the same phenomenon). This figure is redundant with data in the supplementary table.

Author Response

June 16, 2019

To: Danica Jevtic

Associate Editor,

Cancers

Dear Dr. Danica Jevtic,

We submit to your attention the revised version of our manuscript entitled “Elevated Lactate Dehydrogenase Has Prognostic Relevance in Treatment-Naïve Patients Affected by Chronic Lymphocytic Leukemia with Trisomy 12”.

We appreciate the time, dedication and constructive criticism each reviewer provided.

A point-by-point response to reviewers’ comments is outlined below.

Reviewer: 2

Comments and Suggestions for Authors

In this manuscript Autore et al. performed a retrospective analysis of a large cohort of treatment-naïve +12 CLL patient data obtained from multiple Italian centers. They aimed to determine whether specific clinical features of +12 CLL patients were associated with a poor prognosis according to progression-free survival, treatment-free survival, overall survival and CLL-specific survival. Using Cox regression, multivariate analysis revealed that elevated beta-2-microglubulin and LDH were significant risk factors for poor prognosis in all four categories. Beta-2-microgobulin levels are known to be associated with CLL disease burden, and the authors focused on the novelty of identifying LDH as a prognostic indicator for +12 CLL patients. The authors analyzed clinical data from a separate American cohort, and similarly found that elevated LDH was associated with poor prognosis for +12 CLL patients. Surprisingly, they did not identify elevated LDH as a prognostic indicator for negative FISH CLL patients. Overall this manuscript is well written and would be of interest to the CLL field. The following comments might help improve its quality.

The axes of the Kaplan-Meier plot should have a more descriptive label (time is months).

ANSWER. These have been added.

In the Discussion, the authors to should speculate on the mechanism and relevance to the disease of high LDH: Without normal underlying signatures, why is LDH elevated? If elevated LDH is a poor prognosis indicator for +12 CLL, why is it not so for negative FISH CLL?

ANSWER. We think that CLL patients with +12 are characterized by unique clinical and biological features that could explain a high rate of LDH levels above the limit; a possible explanation could be linked also to atypical morphological features of their cells.This has been added to the manuscript.

Is the Forest plot in Figure 4 necessary/appropriate. These plots are normally used to compare data from similar studies (studies examining the same phenomenon). This figure is redundant with data in the supplementary table.

ANSWER. We agree with the reviewer and removed Figure 4.

We also performed general English editing and provided the correct email for one of the co-authors.

We hope that the revised version will be considered acceptable for publication in Cancers.

Please, do not hesitate to contact us should further modifications be required.

Sincerely,

Luca Laurenti